

# Merging ground-based sunshine duration with satellite cloud and aerosol data to produce high resolution long-term surface solar radiation over China

Fei Feng[1] † and Kaicun Wang[2] †

1. College of Forestry, Beijing Forestry University, Beijing 100083, China

2. State Key Laboratory of Earth Surface Processes and Resource Ecology, College of

Global Change and Earth System Science, Beijing Normal University, Beijing, 100875,

China

†These authors contributed equally to this work

**Corresponding Author**:

Fei Feng, College of Forestry, Beijing Forestry University, Email:

forgetbear@bjfu.edu.cn;

Kaicun Wang, College of Global Change and Earth System Science, Beijing Normal

University. Email: kcwang@bnu.edu.cn; Tel: +086 10-58803143; Fax: +086 10-

58800059.



# Abstract

Although great progress has been made in estimating surface solar radiation ($R_s$)
from meteorological observations, satellite retrieval and reanalysis, getting best
estimated of long-term variations in $R_s$ are sorely needed for climate studies. It has been
shown that sunshine duration (SunDu)-derived $R_s$ data can provide reliable long-term
$R_s$ variation. Here, we merge SunDu-derived $R_s$ data with satellite-derived cloud
fraction and aerosol optical depth (AOD) data to generate high spatial resolution (0.1°)
$R_s$ over China from 2000 to 2017. The geographically weighted regression (GWR) and
ordinary least squares regression (OLS) merging methods are compared, and GWR is
found to perform better. Whether or not AOD is taken as input data makes little
difference for the GWR merging results. Based on the SunDu-derived $R_s$ from 97
meteorological observation stations, the GWR incorporated with satellite cloud fraction
and AOD data produces monthly $R_s$ with $R^2 = 0.97$ and standard deviation = 11.14
W/m², while GWR driven by only cloud fraction produces similar results with $R^2 =$
0.97 and standard deviation = 11.41 w/m². This similarity is because SunDu-derived $R_s$
has included the impact of aerosols. This finding can help to build long-term $R_s$
variations based on cloud data, such as Advanced Very High Resolution Radiometer
(AVHRR) cloud retrievals, especially before 2000, when satellite AOD retrievals are
not unavailable. The merged $R_s$ product at a spatial resolution of 0.1° in this study can
be downloaded at https://doi.pangaea.de/10.1594/PANGAEA.921847 (Feng and Wang,

2020).








**Keywords:** surface solar radiation; data fusion; cloud fraction; AOD
**Key Points:**
**(1)** We merge SunDu-derived $R_s$ data with cloud fraction and AOD data to generate
high spatial resolution (0.1°) $R_s$ over China from 2000 to 2017.
**(2)** Whether or not AOD is taken as inputs makes little difference for the GWR merging
results because the SunDu-derived $R_s$ have included the AOD's impact.



## 1. Introduction

A clear knowledge of variations in surface solar radiation ($R_s$) is vitally important for an improved understanding of the global climate system and its interaction with human activity (Jia et al., 2013; Myers, 2005; Schwarz et al., 2020; Wang and Dickinson, 2013; Wild, 2009, 2017; Zell et al., 2015). Widespread direct measurements have shown that $R_s$ has significant decadal variability, namely, a decrease (global dimming) from the 1950s to the late 1980s and subsequent increase (global brightening) (Wild, 2009). The variation in $R_s$ is closely related to Earth's water cycle, the whole biosphere, and the amount of available solar energy. This situation emphasizes the urgency to develop reliable $R_s$ products to obtain the variability in $R_s$.

Great progress has been made in the detection of variability in $R_s$ by meteorological observations, satellite retrieval and radiation transfer model simulations or reanalysis $R_s$ products in previous studies (Rahman and Zhang, 2019; Wang et al., 2015). However, each estimation has its advantages and disadvantages. Direct observed data provide accurate $R_s$ records; however, careful calibration and instrument maintenance are needed. Previous studies have reported that direct observed $R_s$ measurements over China may have major inhomogeneity problems due to sensitivity drift and instrument replacement (Wang, 2014a; Wang et al., 2015; Yang et al., 2018). Before 1990, the imitations of the USSR pyranometers had different degradation rates of the thermopile, resulting in an important sensitivity drift. To overcome radiometer ageing, China replaced its instruments from 1990 to 1993. However, the new solar trackers failed frequently and introduced a high missing data rate for the direct radiation component of $R_s$ (Lu and Bian, 2012; Mo et al., 2008). After 1993, although the instruments were substantially improved, the Chinese-developed pyranometers still had high thermal offset with directional response errors, and the stability of these



instruments was also worse than that of the World Meteorological Organization (WMO)
recommended first-class pyranometers (Lu et al., 2002; Lu and Bian, 2012; Yang et al.,

2010).

Sunshine duration observations collected at weather stations in China have been

used to reconstruct long-term $R_s$ (Feng et al., 2019; He and Wang, 2020; He et al., 2018;
Yang et al., 2020). Based on the global SunDu-derived $R_s$ records, He et al. (2018)
found that SunDu permitted a revisit of global dimming from the 1950s to the 1980s
over China, Europe, and the USA, with brightening from 1980 to 2009 in Europe and
a declining trend $R_s$ from 1994 to 2010 in China. Wang et al. (2015) also found that the
dimming trend from 1961 to 1990 and nearly constant zero trend after 1990 over China,
as calculated from the SunDu-derived $R_s$, was consistent with independent estimates of
AOD (Luo et al., 2001); they also observed changes in the diurnal temperature range
(Wang and Dickinson, 2013; Wang et al., 2012a) and the observed pan evaporation
(Yang et al., 2015). Although direct observations and SunDu-derived $R_s$ can provide
accurate long-term variations in $R_s$, both direct observations and sunshine duration
records are often sparsely spatially distributed.

Satellite $R_s$ retrieval and radiation transfer model simulation or reanalysis $R_s$

products can provide $R_s$ estimation with large spatial coverage. Model simulations and
reanalysis $R_s$ products have substantial biases due to the deficiency of simulating cloud
and aerosol quantities (Feng and Wang, 2019; Zhao et al., 2013). Previous comparative
studies have shown that the accuracies of $R_s$ from reanalyses are lower than those of
satellite products (Wang et al., 2015; Zhang et al., 2016) due to the good capability of
capturing the spatial distribution and dynamic evolution of clouds in satellite remote
sensing data.

**Table 1** lists the current satellite-based $R_s$ products, which have been widely



validated in previous studies. Zhang et al. (2004) found that the monthly International
Satellite Cloud Climatology Project-Flux Data (ISCCP-FD) $R_s$ product had a positive
bias of 8.8 w/m$^2$ using Global Energy Balance Archive (GEBA) archived data as a
reference. By comparing 1151 global sites, Zhang et al. (2015) evaluated four satellite-
based $R_s$ products, including ISCCP-FD, the Global Energy and Water Cycle
Experiment-Surface Radiation Budget (GEWEX-SRB), the University of
Maryland/Shortwave Radiation Budget (UMD-SRB) and the Earth's Radiant Energy
System energy balanced and filled product (CERES EBAF), and concluded that CERES
EBAF shows better agreement with observations than other products. A similar overall
good performance of CERES EBAF can also be found (Feng and Wang, 2018; Ma et
al., 2015).
**Table 1**. Current satellite-derived surface solar radiation ($R_s$) products

| Satellite $R_s$ product | Source | Spatial resolution | Time range |
|---|---|---|---|
| ISCCP-FD | ISCCP | 2.5° | 1983-2009 |
| GEWEX-SRB | ISCCP-DX | 1° | 1983-2007 |
| UMD-SRB | METEOSAT-5 | 0.5° | 1983-2007 |
| GLASS-DSR | Terra/Aqua, GOES, MSG, MTSAT | 0.05° | 2008-2010 |
| CLARA-A2 | AVHRR | 0.25° | 1982-2015 |
| MCD18A1 | Terra/Aqua, MODIS | 5.6 km | 2001-present |
| Himawari-8 SWSR | Himawari-8 | 5 km | 2015-present |
| SSR-tang | ISCCP-HXG, ERA5, MODIS | 10 km | 1982-2017 |
| Cloud_cci AVHRR-PMv3 | AVHRR/CC4CL | 0.05° | 1982-2016 |


Although CERES EBAF uses more accurate input data to provide $R_s$ data, its
spatial resolution is only 1° (Kato et al., 2018). Since 2010, new-generation
geostationary satellites have provided opportunities for high temporal and spatial
resolution $R_s$ data, such as Himawari-8 (Hongrong et al., 2018; Letu et al., 2020).
However, the time span of the new-generation satellite-based $R_s$ product is short. The
long-term AVHRR records provide the possibility of building long-term radiation



datasets. The CLoud, Albedo and RAdiation dataset, the AVHRR-based data-second
edition (CLARA-A2), covers a long time period, but the spatial resolution is only 0.25 °
(Karlsson et al., 2017). Recently, Tang et al. (2019) built a satellite-based $R_s$ (SSR-tang)
dataset using ISCCP-HXG cloud data. By using a variety of cloud properties derived
from AVHRR, Stengel et al. (2020) presented the Cloud_cci AVHRR-PMv3 dataset
generated within the Cloud_cci project. However, the long-term cloud records also
contain uncertainties. For example, ISCCP cloud products, which directly combine
geostationary and polar orbiter satellite-based cloud data, have large inhomogeneity due
to different amounts of data from polar orbit and geostationary satellites and their
different capabilities for detecting low-level clouds (Dai et al., 2006; Evan et al., 2007).
This inhomogeneity of the cloud products might introduce significant inhomogeneity
to the $R_s$ values calculated from the cloud products (Montero-Martń et al., 2020;
Pfeifroth et al., 2018b), and $R_s$ long-term variability estimation still needs improvement.

Efforts have been made to further improve $R_s$ products. Merging multisource data

has become an effective empirical method for improving the quality of $R_s$ products
(Camargo and Dorner, 2016; Feng and Wang, 2018; Hakuba et al., 2014; Journé et al.,
2012; Lorenzo et al., 2017; Ruiz-Arias et al., 2015). For instance, to produce
spatiotemporally consistent $R_s$ data, multisource satellite data are used in Global LAnd
Surface Satellite (GLASS) $R_s$ products (Jin et al., 2013). By merging reanalysis and
satellite $R_s$ data by the probability density function-based method, the reanalysis $R_s$
biases can be substantially reduced (Feng and Wang, 2018). This finding suggests that
fusion methods are effective ways to improve the estimation of $R_s$, especially when $R_s$
impact factors are considered (Feng and Wang, 2019). Although linear regression
fusion methods can produce $R_s$ data incorporated with $R_s$ impact factors, the stable
regression parameters might have negative effects on the final fusion results due to the





complex characteristics of $R_s$ spatial-temporal variability.
On the other hand, the spatial resolution of $R_s$ data is crucial for regional
meteorology studies, as the minimum requirement of the spatial resolution of $R_s$ data,
as suggested by the Observing Systems Capabilities Analysis and Review of WMO
OSCAR), is 20 km (Huang et al., 2019). Interpolation methods are often included in $R_s$
fusion methods to further improve the spatial resolutions of $R_s$ data (Loghmari et al.,
2018). For example, Zou et al. (2016) estimated global solar radiation using an artificial
neural network based on an interpolation technique in southeast China. By integrating
$R_s$ data from 13 ground stations with Meteosat Second Generation satellite $R_s$ products,
Journée and Bertrand (2010) found that kriging with the external drift interpolation
method performed better than mean bias correction, interpolated bias correction and
ordinary kriging with satellite-based correction. However, interpolation results have
uncertainties due to the lack of detailed high spatial resolution information. Although
traditional linear regression fusion methods can incorporate high spatial resolution data
during the fusion process, the impacts of the stable regression parameters need further
investigation.
Geographically weighted regression (GWR) is an extension of the traditional
regression model by allowing the relationships between dependent and explanatory
variables to vary spatially. Researchers have examined and compared the applicability
of GWR for the analysis of spatial data relative to that of other regression methods (Ali
et al., 2007; Gao et al., 2006; Georganos et al., 2017; LeSage, 2004; Sheehan et al.,
2012; Zhou et al., 2019a). Due to the large spatial heterogeneity of $R_s$ over China, the
GWR method might produce accurate $R_s$ variability estimations with an improved
spatial resolution.
This study is established to merge SunDu-derived $R_s$ data with satellite-derived



cloud fraction (CF) and AOD data to generate high spatial resolution (0.1) $R_s$ over China
from 2000 to 2017. The GWR and ordinary least squares (OLS) regression merging
methods are compared. CF and AOD are important $R_s$ impact factors. In this study,
whether much improvement is made in merging $R_s$ by incorporating AOD is also
evaluated. The output of this study can provide guidance to merge multisource data to
generate long-term $R_s$ data over China. Direct $R_s$ observations and sunDu data records
from CMDC cannot be easily downloaded for the researchers from outside China due
to the authentication of the China data use policy. This further demonstrate the
importance of our merged $R_s$ product.

## 2. Data and Methodology

### 2.1. Ground-based observations
### 2.2.1 Direct observations

$R_s$ direct observations from 2000 to 2016 are obtained from the China
Meteorological Data Service Center (CMDC, http://data/cma/cn/) of the China
Meteorological Administration (CMA). TBQ-2 pyranometers and DFY4 pyranometers
have been used to measure $R_s$ since 1993. Daily $R_s$ values from 97 $R_s$ stations are
collected, and we calculated monthly $R_s$ values by averaging daily $R_s$ values when daily
observed data are available for more than 15 days for each month at each radiation
station. These monthly $R_s$ values from direct measurements and collocated SunDu-
derived $R_s$ are used as independent reference data to investigate the performances of the
fusion methods (**Fig. 1**). The whole area over China is further divided into nine zones
by the K-mean cluster method based on geographic locations and multiyear mean $R_s$
using 97 $R_s$ direct observation sites, as shown in **Figure 1**. The download instructions
of the $R_s$ direct observations can be found in **table 2**.

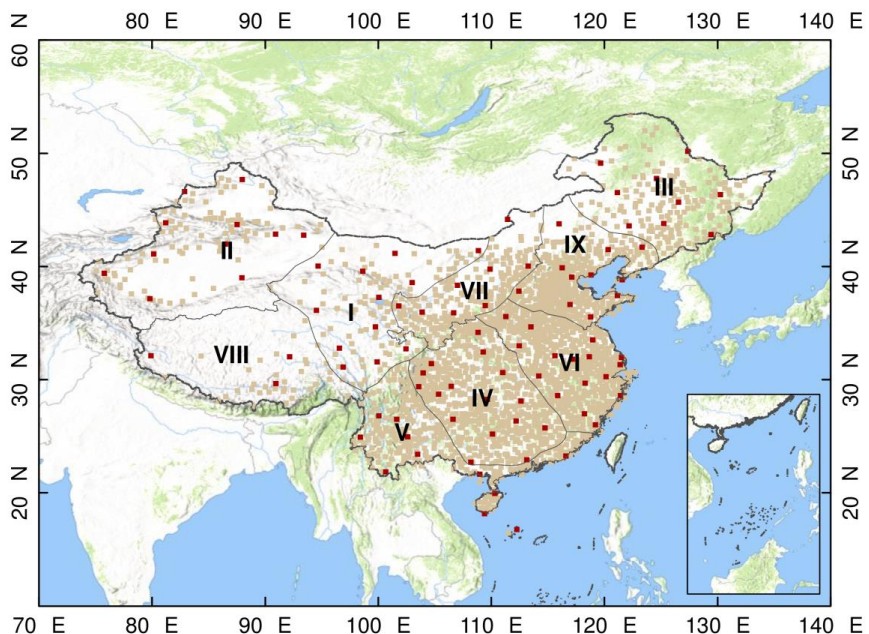

**Figure 1.** The 2,400 sunshine duration (SunDu) merging sites are shown as light brown points, and 97 independent validation sites, including $R_s$ direct measurements and SunDu-derived $R_s$ measurements, are shown as brown red points. The whole region is classified into nine subregions (I to IX) by the K-mean cluster method based on geographic locations and multiyear mean $R_s$ using 97 $R_s$ direct observation sites. The base hillshade map was produced by an elevation map of China using the global digital elevation model (DEM) derived from the Shuttle Radar Topography Mission 30 (SRTM30) dataset.

**Table 2.** Summary of availability information for all source data used in this study.
CMDC is the China Meteorological Data Service Center. SunDu is the sunshine
duration data. $R_s$ is surface solar radiation and AOD is the aerosols optical depth.

| Data Source | Derived Parameters used in this Study | Version Number | Access Point | Notes |
|---|---|---|---|---|
| Direct $R_s$ measurement data from CMDC | $R_s$ | Version 1.0 | http://data/cma/cn/ | Authentication is required for the China data use policy |
| SunDu observations and other meteorological data | $R_s$ | Version 1.0 | http://data/cma/cn/ | Authentication is required for the China data use policy |
| CERES EBAF | $R_s$ | Ed4.1 | https://ceres.larc.nasa.gov/data/#ebaf-level-3b | A email address to order the data |
| CERES SYN1deg | AOD | Ed4A | https://ceres.larc.nasa.gov/data/#syn1deg-level-3 | A email address to order the data |
| MODAL2 M CLD | cloud fraction | - | https://neo.sci.gsfc.nasa.gov/view.php?datasetId=MODAL2_M_CLD_FR | Directly download |


### 2.2.2 SunDu-derived Rs observations

Sunshine duration observations (SunDu) and other meteorological data (e.g., air
temperature, relative humidity and surface pressure) from 1980 to 2017, which were
collected from approximately 2,400 meteorological stations (http://data/cma/cn/) from
the CMA, are used to calculate the SunDu-derived $R_s$ (**Fig. 1**). $R_s$ values are calculated
following the method of the revised Ångström-Prescott equation (Eq. (1-2)) (He et al.,
2018; Wang, 2014a; Wang et al., 2015; Yang et al., 2006).
$$\frac{R_s}{R_c} = a_0 + a_1 \frac{n}{K} + a_2 (\frac{n}{K})^2 \qquad (1)$$
$$R_c = \int (\tau_{c\_dir} + \tau_{c\_dif}) \times I_0 d_t \qquad (2)$$
where n represents the measured SunDu, and K represents the theoretical value of the

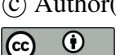



SunDu. $a_0$, $a_1$, and $a_2$ are the station-dependent parameters (Wang, 2014a). $R_c$ is the
daily total solar radiation at the surface under clear-sky conditions (Eq. 2). $\tau_{c\_dir}$ and $\tau_{c\_dif}$
represent the direct radiation transmittance and the diffuse radiation transmittance under
clear-sky conditions. $I_0$ is the solar irradiance at the top of the atmosphere (TOA).

SunDu data are relatively widely distributed and have a long-term record

(Sanchez-Lorenzo et al., 2009; Wild, 2009). Existing studies have also confirmed that
SunDu-derived $R_s$ data are reliable $R_s$ data, which can capture long-term trends of $R_s$
and reflect the impacts of both aerosols and clouds at time scales ranging from daily to
decadal (Feng and Wang, 2019; Manara et al., 2015; Sanchez-Lorenzo et al., 2013;
Sanchezromero et al., 2014; Tang et al., 2011; Wang et al., 2012b; Wild, 2016).

Based on the classified subregions using 97 direct $R_s$ observations in **Figure 1**, the

intercomparison results in **Figure 2** and **Figure 3** show that the agreement between
SunDu-derived $R_s$ and CERES EBAF $R_s$ estimates is better than that between the direct
observations and SunDu-derived $R_s$ estimates, which is likely due to the inhomogeneity
issue of direct $R_s$ observations over China, as mentioned in many previous studies
(Wang, 2014b; Yang et al., 2018). These results indicate that SunDu-derived $R_s$ data can
be used to analyse the variation in $R_s$ over China.

The SunDu-derived $R_s$ observations, excluding SunDu observations located at

direct observation sites, are used for merging. Ten percent merging observations are
randomly selected for GWR parameter optimization. The download instructions of the
SunDu observations can be found in **table 2**.

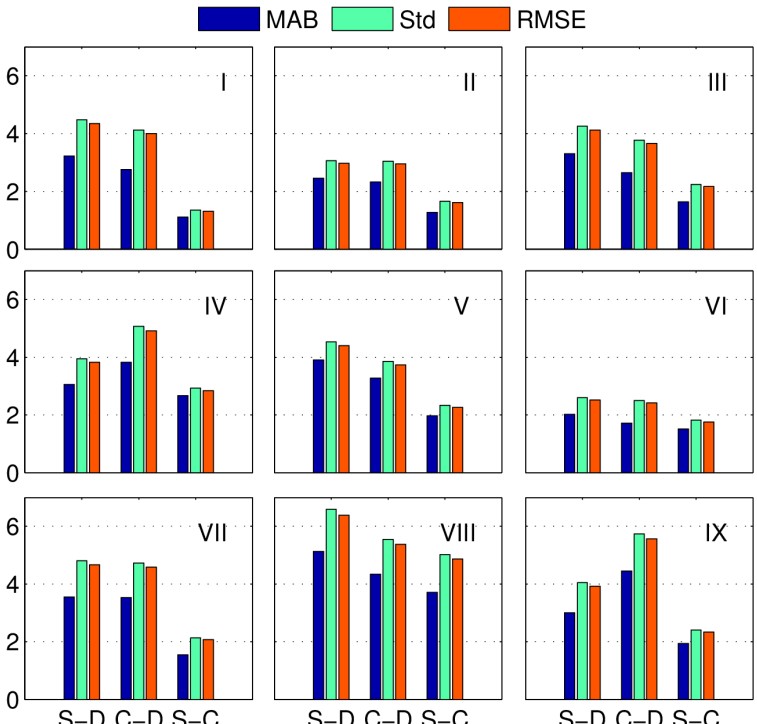

**Figure 2.** Statistical summary of annual anomaly $R_s$ from direct observed $R_s$, SunDu-derived $R_s$ and CERES EBAF $R_s$ estimates in different subregions. The statistics include the mean absolute bias (MAB), standard deviation (Std) and root mean square error (RMSE). We use MAB due to the cancelling out effect of positive bias and negative bias. Nine subregions (I to IX) over China are shown in Figure 1. S-D represent comparisons between SunDu-derived $R_s$ and directly observed $R_s$. C-D represent comparison between CERES EBAF $R_s$ and directly observed $R_s$. S-C represent comparisons between SunDu-derived $R_s$ and CERES EBAF $R_s$.

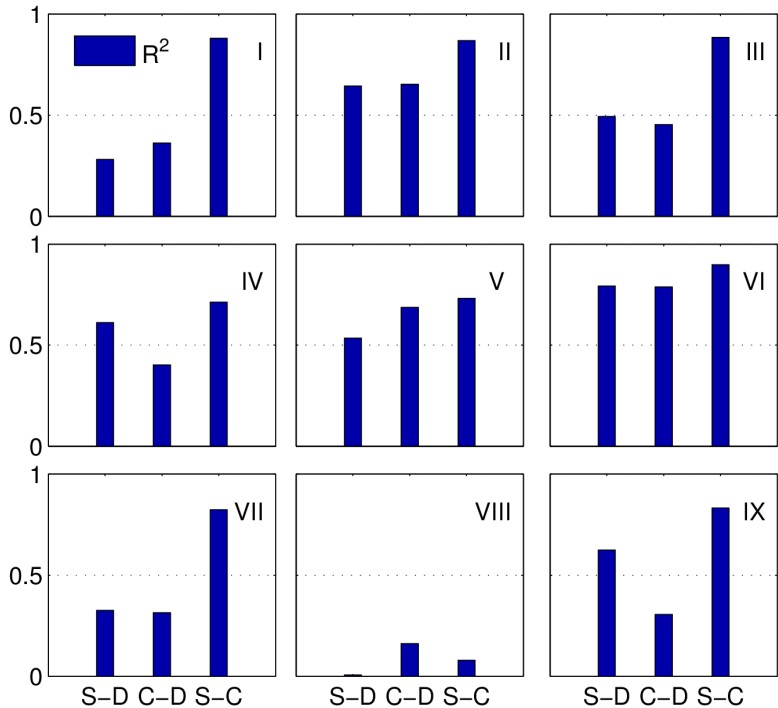

**Figure 3.** Similar to Figure 2, but this statistical summary is for $R^2$.

### 2.2. Satellite data

$R_s$ data from the Clouds and Earth's Radiant Energy System energy balanced and

filled product (CERES Synoptic (CERES) EBAF) surface product (edition 4.1) (Kato

et al., 2018), cloud fraction from MODAL2 M CLD data product (Platnick et al., 2017)

and AOD from the CERES SYN1deg) edition 4A product (Doelling et al., 2013) are

collected in this study. CERES EBAF $R_s$ data are used as reference data. AOD from

CERES SYN1deg and cloud fraction from MODAL2 M CLD are used as input data for

fusion methods.

CERES is a 3-channel radiometer measuring three filtered radiances, including

shortwave (0.3-5 μm), total (0.3-200 μm) and window (8-12 μm). $R_s$ from CERES

EBAF are adjusted using radiative kernels, including bias correction and Lagrange



multiplier processes. The input data of CERES EBAF are adjusted during the product
generating process constrained by CERES observations at the TOA. The biases in
temperature and specific humidity from the Goddard Earth Observing System (GEOS)
reanalysis are adjusted by atmospheric infrared sounder (AIRS) data. Cloud properties,
such as optical thickness and emissivity, from MODIS and geostationary satellites are
constrained by cloud profiling radar, Cloud-Aerosol Lidar, and Infrared Pathfinder
Satellite Observations (CALIPSO) detectors and CloudSat. The uncertainties of
CERES EBAF data, reported by (Kato et al., 2018), in all sky global annual mean $R_s$ is
4 W m$^{-2}$. Previous studies (Feng and Wang, 2019; Feng and Wang, 2018; Ma et al.,
2015; Wang et al., 2015) have shown that the CERES EBAF surface product provides
reliable estimations of $R_s$.
CERES SYN1deg AOD derived from an aerosol transport model, named
Atmospheric Transport and Chemistry Modelling (MATCH) (Collins et al., 2001),
which assimilates MODIS AOD data, is used to obtain spatiotemporally consistent
AOD data. Different aerosol constituents, including small dust (<0.5 μm), large dust
(>0.5 μm), stratosphere, sea salt, soot and soluble, are used to compute the optical
thickness for a given constituent optical thickness for a given constituent.
Cloud fraction data from MODAL2 M CLD are collected as input cloud fraction
data with a spatial resolution of 0.1 ° and time span from 2000 to 2017 (Platnick et al.,
2017). The MODAL2 M CLD data are synthesized based on the cloud data from
MOD06. Cloud fraction data from MOD06 are generated by the cloud mask product of
MOD35 with a spatial resolution of 1 km. The MOD35 cloud mask is determined by
applying appropriate single field of view (FOV) spectral tests to each pixel with a series
of visible and infrared threshold and consistency tests. Each land type has different
algorithms and thresholds for the tests. For each pixel test, an individual confidence





flag is determined and then combined to produce the final cloud mask flag. The three
confidence levels included in the cloud mask flag output are (i) high confidence for
cloudless pixels (Group confidence values > 0.95); (ii) low confidence for unobstructed
views on the surface (Group confidence values $Q \leq 0.66$); and (iii) values between 0.66
and 0.95, and spatial and temporal continuity tests are further applied to determine
whether the pixel is absolutely cloudless. Then, the cloud fraction is calculated from
the 5 x 5-km cloud mask pixel groupings, i.e., given the 25 pixels in the group, the
cloud fraction for the group equals the number of cloudy pixels divided by 25.
***2.3. Methods***
**2.3.1 Fusion models**
OLS regression and GWR are used to build fusion methods for estimating $R_s$ data.
Clouds fraction and AOD have been important factors that affect variations in $R_s$. We
compare different combinations of input data for the fusion methods, which can be
classified into two types. The first type only contains cloud fraction data. The second
type contains clouds fraction and AOD (Feng and Wang, 2020).
GWR is a regression model that allows the relationships between the independent
and dependent variables to vary by locality (Brunsdon et al., 2010; Brunsdon et al.,
1998). GWR deviates from the assumption of homoskedasticity or static variance but
calculates a specific variance for data within a zone or search radius of each predictor
variable (Brunsdon et al., 1998; Fotheringham et al., 1996; Sheehan et al., 2012). The
regression coefficients in GWR are not based on global information; rather, they vary
with location, which is generated by a local regression estimation using subsampled
data from the nearest neighbouring observations. The principle of GWR is described as
follows:

$$y_i = \delta(i) + \sum_k \delta_k(i)x_{ik} + \varepsilon_i \qquad (3)$$





where $y_i$ is the value of $R_s$ unit $i$; $i=1,2,...,n$, $n$ denotes location $i$, $x_{ik}$ indicates the value
of the $x_{ik}$ variable, such as cloud fraction and AOD, and $\varepsilon$ denotes the residuals. $\delta_{(i)}$ is
the regression intercept. $\delta_{k(i)}$ is the vector of regression coefficients determined by
spatial weighting function $w_{(i)}$, which is the weighting function quantifying the
proximities of location $i$ to its neighbouring observation sites; $X$ is the variable matrix,
and $b$ is the bias vector.

$$\delta_k(i) = (X^T w(i) X)^{-1} X^T w(i) b \qquad (4)$$

The weighting functions are generally determined using the threshold method,
inverse distance method, Gauss function method, and Bi-square method. Due to the
irregular distribution of observation sites and computer ability, the adaptive Gaussian
function method is selected as a weighting function that varies in extent as a function
of $R_s$ observation site density.

$$w_{ij} = \exp(-(d_{ij}/b)^2) \qquad (5)$$

where $w_{ij}$ is the weighting function for observation site $j$ that refers to location $i$; $d_{ij}$
denotes the Euclidian distance between $j$ and $i$; and b is the size of the neighbourhood,
the maximum distance away from regression location $i$, called "bandwidth", which is
determined by the number of nearest neighbour points (NNPs).
**2.3.2 GWR parameter comparison**
To perform the local regression for every local area, the numbers of NNPs are
required to estimate spatially varying relationships between CF, AOD and $R_s$ in the
GWR-based fused method. To identify the best combination of parameter values, we
test the numbers of NNPs ranging from 29 to 1000. Ten percent of merging SunDu-
derived $R_s$ data are randomly selected to validate these GWR parameters (**Fig. 1**). The
results show that $R^2$ increases and bias decreases when the number of NNPs decreases.
However, when the NNP is smaller than 30, the GWR-based fusion method produces





spatially incomplete $R_s$ data due to the local collinearity problem with large spatial
variability. Therefore, 30 is selected as the NNP parameter (**Table 3**).

**Table 3.** Statistical summary of GWR parameter optimization. NPP is the number of
nearest neighbour points. GWR-CF presents the GWR-based fused method using only
cloud fraction (CF) input, and GWR-CF-AOD presents that of using both CF and
aerosol optical depth (AOD) as input. MAB is the mean absolute bias. Std is the
standard deviation. RMSE is the root mean square error.

| NNP | GWR-CF | | | | | GWR-CF-AOD | | | | |
|---|---|---|---|---|---|---|---|---|---|---|
| | $R^2$ | Bias | MAB | Std | RMSE | $R^2$ | Bias | MAB | Std | RMSE |
| 29 | 0.91 | -0.21 | 7.45 | 9.90 | 9.90 | 0.91 | -0.13 | 7.47 | 9.93 | 9.92 |
| 30 | 0.91 | -0.23 | 7.45 | 9.90 | 9.90 | 0.91 | -0.14 | 7.47 | 9.92 | 9.91 |
| 31 | 0.91 | -0.24 | 7.45 | 9.90 | 9.90 | 0.91 | -0.14 | 7.47 | 9.91 | 9.91 |
| 32 | 0.91 | -0.25 | 7.46 | 9.91 | 9.91 | 0.91 | -0.14 | 7.47 | 9.91 | 9.90 |
| 33 | 0.91 | -0.26 | 7.47 | 9.92 | 9.92 | 0.91 | -0.15 | 7.46 | 9.90 | 9.90 |
| 34 | 0.91 | -0.27 | 7.47 | 9.93 | 9.93 | 0.91 | -0.14 | 7.46 | 9.90 | 9.89 |
| 35 | 0.91 | -0.28 | 7.48 | 9.94 | 9.94 | 0.91 | -0.14 | 7.46 | 9.89 | 9.88 |
| 36 | 0.91 | -0.28 | 7.49 | 9.94 | 9.94 | 0.91 | -0.14 | 7.46 | 9.89 | 9.88 |
| 37 | 0.91 | -0.29 | 7.49 | 9.95 | 9.95 | 0.91 | -0.14 | 7.46 | 9.88 | 9.87 |
| 38 | 0.91 | -0.30 | 7.50 | 9.96 | 9.96 | 0.91 | -0.14 | 7.46 | 9.88 | 9.87 |
| 39 | 0.91 | -0.31 | 7.51 | 9.98 | 9.98 | 0.91 | -0.14 | 7.46 | 9.87 | 9.87 |
| 40 | 0.91 | -0.32 | 7.52 | 9.99 | 9.99 | 0.91 | -0.14 | 7.46 | 9.87 | 9.87 |
| 50 | 0.90 | -0.38 | 7.62 | 10.12 | 10.12 | 0.91 | -0.12 | 7.51 | 9.91 | 9.91 |
| 100 | 0.89 | -0.57 | 8.20 | 10.90 | 10.91 | 0.90 | -0.02 | 7.86 | 10.31 | 10.30 |
| 500 | 0.81 | -1.08 | 10.89 | 14.50 | 14.54 | 0.86 | 0.20 | 9.55 | 12.45 | 12.45 |
| 1000 | 0.75 | -1.13 | 12.60 | 16.57 | 16.61 | 0.82 | 0.26 | 10.68 | 13.84 | 13.85 |


## 359  3. Results

### 360  3.1 Site validation

Based on the independent SunDu validation sites, both the GWR and OLS
methods explain 97%~86% of $R_s$ variability (**Fig. 4**). The GWR method generally
shows an improved performance compared with the OLS method due to the
representativeness of the spatial heterogeneity relationship between $R_s$ and its impact
factors in GWR. Both the GWR and OLS methods produce better simulations of $R_s$ if



satellite and AOD data are incorporated.
Direct observations from 2000 to 2016 are also used to further evaluate the
performance of the fusion methods (**Fig. 4**). The comparative result shows that both
fusion methods show slightly reduced performances when using direct $R_s$ observations
rather than the SunDu-derived $R_s$. Both the GWR and OLS methods explain 91%~82%
of $R_s$ variability by using direct observations as reference data. Similarly, the GWR
method exhibits better performances than the OLS-based fusion method, with an $R^2$ of
0.91 and root mean square error (RMSE) ranging from 19.89 to 19.97 W/m$^2$ at the
monthly time scale (**Table 4**).

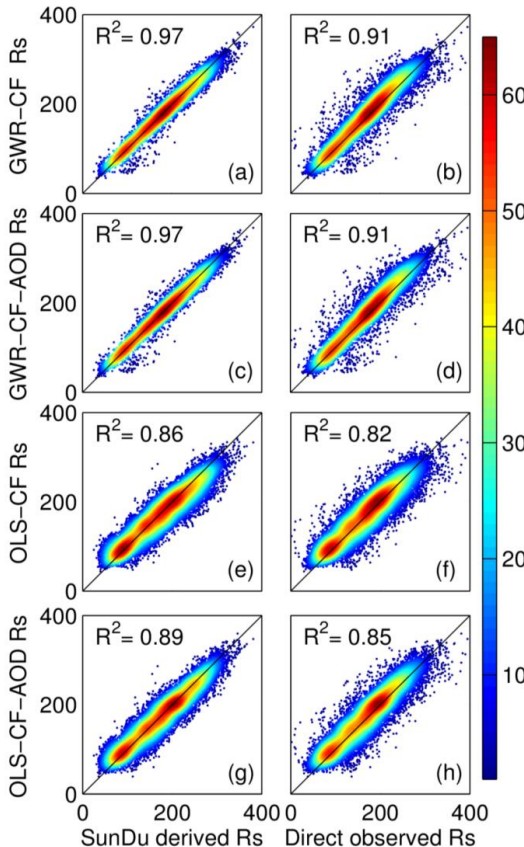


**Figure 4.** Comparison of surface solar radiation ($R_s$) derived from the GWR method



and the OLS method. Subplots (a, c, e, g) represent validation results using SunDu-
derived $R_s$ data as a reference, while that of subplots (b, d, f, h) use directly observed
$R_s$ data. Subplots (a, b, c, d) denote the GWR validation results, and subplots (e, f, g, h)
denote the OLS validation results.

**Table 4.** Validation of fusion methods driven by cloud fraction (CF) and AOD. GWR-
CF and OLS-CF represent the GWR fusion method and OLS fusion method driven only
by CF. GWR-CF-AOD and OLS-CF-AOD represent GWR and OLS fusion methods
driven by CF and AOD, respectively.

|  | Time scale | Ref | R2 | Bias | Std | RMSE |
|---|---|---|---|---|---|---|
| GWR-CF | monthly | SunDu $R_s$ | 0.97 | -1.17 | 11.41 | 11.47 |
| GWR-CF-AOD | monthly | SunDu $R_s$ | 0.97 | -0.82 | 11.14 | 11.17 |
| OLS-CF | monthly | SunDu $R_s$ | 0.86 | -3.80 | 25.03 | 25.32 |
| OLS-CF-AOD | monthly | SunDu $R_s$ | 0.89 | -1.37 | 22.10 | 22.15 |
| GWR-CF | monthly | Direct Obs | 0.91 | 4.88 | 19.29 | 19.89 |
| GWR-CF-AOD | monthly | Direct Obs | 0.91 | 5.24 | 19.27 | 19.97 |
| OLS-CF | monthly | Direct Obs | 0.82 | 2.18 | 26.73 | 26.82 |
| OLS-CF-AOD | monthly | Direct Obs | 0.85 | 4.64 | 24.71 | 25.15 |
| GWR-CF | spring | SunDu $R_s$ | 0.95 | -1.3 | 11.5 | 11.57 |
| GWR-CF-AOD | spring | SunDu $R_s$ | 0.95 | -0.86 | 11.2 | 11.23 |
| OLS-CF | spring | SunDu $R_s$ | 0.77 | -4.97 | 23.65 | 24.16 |
| OLS-CF-AOD | spring | SunDu $R_s$ | 0.84 | -1.35 | 19.85 | 19.9 |
| GWR-CF | summer | SunDu $R_s$ | 0.9 | -2.09 | 14.08 | 14.23 |
| GWR-CF-AOD | summer | SunDu $R_s$ | 0.9 | -1.38 | 13.76 | 13.82 |
| OLS-CF | summer | SunDu $R_s$ | 0.65 | -6.49 | 26.18 | 26.97 |
| OLS-CF-AOD | summer | SunDu $R_s$ | 0.77 | -1.37 | 21.17 | 21.22 |
| GWR-CF | autumn | SunDu $R_s$ | 0.95 | -1.27 | 9.48 | 9.56 |
| GWR-CF-AOD | autumn | SunDu $R_s$ | 0.96 | -1.04 | 9.17 | 9.23 |
| OLS-CF | autumn | SunDu $R_s$ | 0.67 | -3.22 | 25.62 | 25.82 |
| OLS-CF-AOD | autumn | SunDu $R_s$ | 0.71 | -1.97 | 23.79 | 23.87 |
| GWR-CF | winter | SunDu $R_s$ | 0.94 | 0.01 | 9.87 | 9.86 |
| GWR-CF-AOD | winter | SunDu $R_s$ | 0.94 | 0.04 | 9.78 | 9.78 |
| OLS-CF | winter | SunDu $R_s$ | 0.63 | -0.37 | 24.16 | 24.16 |
| OLS-CF-AOD | winter | SunDu $R_s$ | 0.65 | -0.78 | 23.41 | 23.42 |
| GWR-CF | annual | Direct Obs | 0.37 | 5.62 | 4.73 | 10.42 |
| GWR-CF-AOD | annual | Direct Obs | 0.37 | 5.98 | 4.79 | 10.53 |
| OLS-CF | annual | Direct Obs | 0.30 | 3.06 | 5.01 | 15.01 |
| OLS-CF-AOD | annual | Direct Obs | 0.33 | 5.45 | 4.89 | 13.34 |
| GWR-CF | annual | SunDu $R_s$ | 0.57 | -1.19 | 4.30 | 6.76 |
| GWR-CF-AOD | annual | SunDu $R_s$ | 0.58 | -0.84 | 4.30 | 6.68 |
| OLS-CF | annual | SunDu $R_s$ | 0.35 | -3.58 | 5.63 | 15.17 |





| | | | | | | | |
|---|---|---|---|---|---|---|---|
| OLS-CF-AOD | annual | SunDu $R_s$ | 0.39 | -1.23 | 5.44 | 13.40 |
| GWR-CF | annual mean | SunDu $R_s$ | 0.94 | -1.50 | 6.63 | 6.76 |
| GWR-CF-AOD | annual mean | SunDu $R_s$ | 0.95 | -1.15 | 6.41 | 6.47 |
| OLS-CF | annual mean | SunDu $R_s$ | 0.62 | -3.90 | 17.11 | 17.46 |
| OLS-CF-AOD | annual mean | SunDu $R_s$ | 0.71 | -1.58 | 14.90 | 14.90 |
| GWR-CF | annual mean | Direct Obs | 0.89 | 5.08 | 9.85 | 11.03 |
| GWR-CF-AOD | annual mean | Direct Obs | 0.89 | 5.43 | 9.75 | 11.11 |
| OLS-CF | annual mean | Direct Obs | 0.70 | 2.57 | 16.31 | 16.42 |
| OLS-CF-AOD | annual mean | Direct Obs | 0.77 | 4.88 | 14.00 | 14.75 |


### 3.2 Seasonal and annual variations in $R_s$

To analyse the impacts of AOD on the GWR fusion results, the GWR driven with
only CF (GWR-CF) and GWR driven with CF and AOD (GWR-CF-AOD) are
compared. Two validation sites (Chang Chun, 43.87 °N 125.33 °E and Bei Hai, 21.72 °N
109.08 °E) are randomly selected to evaluate the seasonal and annual variations in $R_s$
derived from the GWR method (**Fig. 5**). As shown in **subplots (a and b)**, both GWR-
CF and GWR-CF-AOD produce similar seasonal variation patterns compared with
SunDu-derived $R_s$ and CERES EBAF $R_s$ data. Small differences are found in the
seasonal variation in $R_s$ derived from GWR regardless of whether AOD was
incorporated. Examination of the annual variation $R_s$ from the GWR-CF and GWR-CF-
AOD are shown in **subplots (c and d)** of **Figure 5**. The two fusion methods also
produce similar annual $R_s$ variations. The similar performances of the GWR-CF and
GWR-CF-AOD might suggest that the impacts of AOD have already been included in
the SunDu-derived $R_s$ site data.



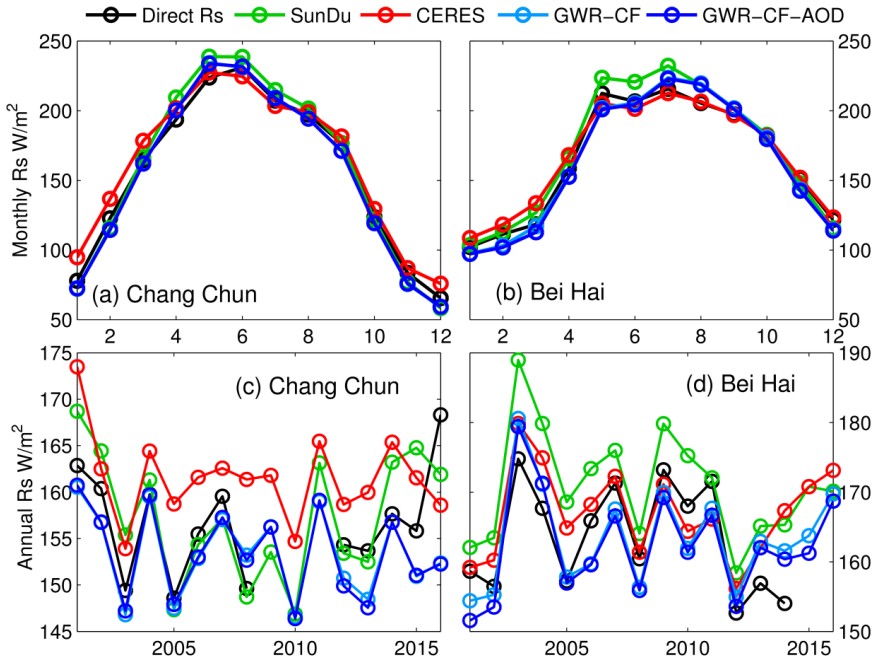

**Figure 5.** Seasonal and annual variations in $R_s$ at two sites: Chang Chun (a and c,
43.87 °N and 125.33 °E) and Bei Hai (b and d, 23.50 °N, 99.72 °E). SunDu $R_s$ is the
SunDu-derived $R_s$ data, and GWR-CF $R_s$ is $R_s$ produced by the GWR method
incorporating only the cloud fraction. GWR-CF-AOD is $R_s$ produced by the GWR
method incorporating cloud fraction and AOD.

We also analysed the performances of fusion methods for different seasons at all
validation sites, as shown in **Table 4**. At seasonal scales, both the GWR-CF and GWR-
CF-AOD methods have high $R^2$ values ranging from 0.94 to 0.96, compared with direct
$R_s$ measurement or SunDu-derived $R_s$. GWR-CF and GWR-CF-AOD show slight
differences, indicating that both fusion methods produce consistent $R_s$ seasonal
variation patterns, which might be because the impacts of AOD have already been
included in the SunDu-derived $R_s$ site data at seasonal time scales. Comparatively, the
GWR methods perform best in autumn, with RMSEs ranging from 9.23W/m$^2$ to 9.56



W/m$^2$ followed by winter, spring and summer. Both the GWR-CF and GWR-CF-AOD
methods produce similar annual variations in $R_s$ from 2000 to 2016, with R$^2$ values
ranging from 0.57 to 0.58 (**Table 4**). The statistics indicate that the GWR can produce
reasonable seasonal and annual variations in $R_s$.
**3.3 Multiyear mean and long-term variability in $R_s$**

**Figure 6** shows the performance of GWR-CF and GWR-CF-AOD on simulating

the multiyear mean $R_s$ by using 97 direct $R_s$ observation sites and independent SunDu-
derived $R_s$ sites. Based on direct $R_s$ measurements, both GWR-based methods show
good performances with high R$^2$ (0.89~0.95) and low RMSE (11.03~11.11 W/m$^2$), and
few differences are found for the GWR merging results, whether or not AOD is taken
as input data (**Table 4**).

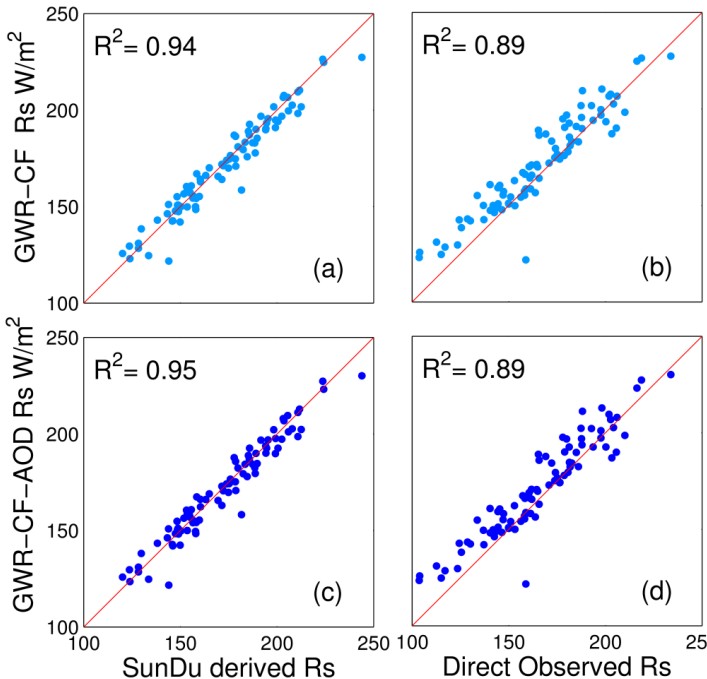


**Figure 6.** Comparison of multiyear mean surface solar radiation ($R_s$) derived from the
GWR method. Subplots (a, c) represent validation results using SunDu-derived $R_s$ data
as a reference, while that of subplots (b, d) use direct observed $R_s$ data.
The spatial distributions of the multiyear means of $R_s$ from 2000 to 2017 are shown
in **Figure 7**. The SunDu sites show that $R_s$ is high in northwest China, ranging from 180
to 300 W/m$^2$, and low in eastern China, ranging from 120 to 180 W/m$^2$. Both the GWR-
CF and GWR-CF-AOD methods show consistent $R_s$ spatial patterns with SunDu-
derived $R_s$ observations and CERES EBAFs, indicating that the relationship between
$R_s$ and impact factors is not linearly stable and is closely related to spatial position. The
spatial distribution of the $R_s$ trend derived from the GWR method is also consistent with
the SunDu-derived $R_s$ trend, especially in western China (**Fig. 8**).

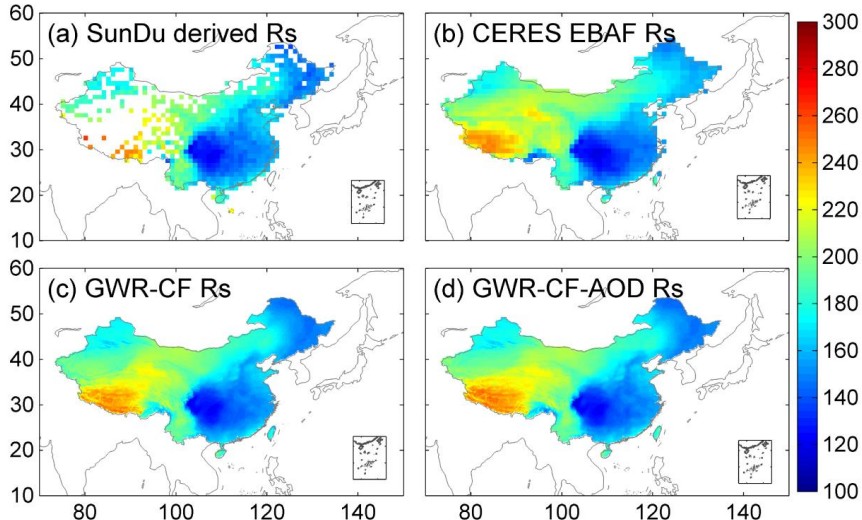


**Figure 7.** Spatial distribution of multiyear mean monthly surface solar radiation ($R_s$)

from 2000 to 2017. The first line (a, b) shows the observed multiyear mean monthly $R_s$
from SunDu and CERES EBAF; the multiyear mean monthly $R_s$ derived from the GWR
method are shown in the second line (c, d), respectively.

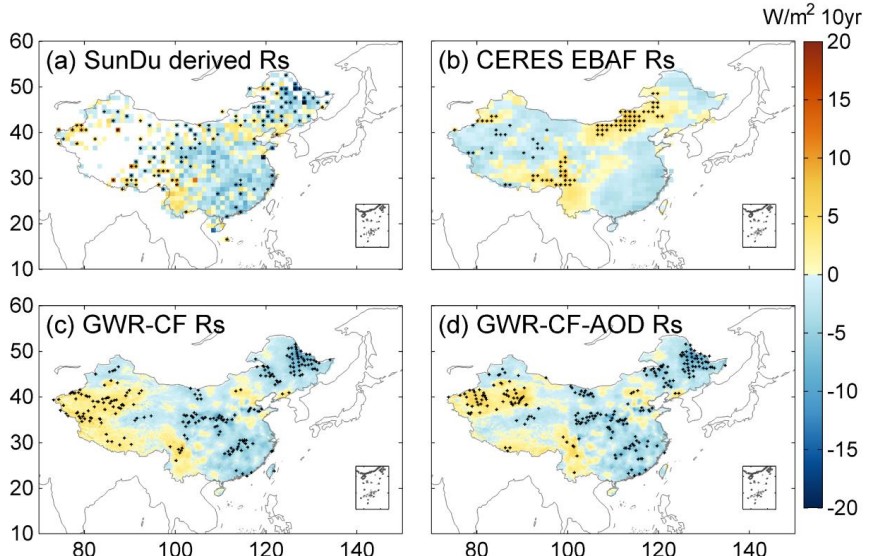

**Figure 8.** Spatial distributions of monthly anomaly trends of surface solar radiation ($R_s$)

from 2000 to 2017. The first line (a, b) shows the SunDu-derived $R_s$ and CERES EBAF

$R_s$; the $R_s$-derived GWR fusion methods are shown in the second line (c, d). Subplots

(c) incorporate only CF, and subplots (d) incorporate CF and AOD. The black dots on

the maps represent significant trends (P<0.05).

Based on the classified subregions using 97 direct $R_s$ observations in **Figure 1**, the

regional means of $R_s$ annual anomaly variation from 2000 to 2016 are shown in **Figure**

**9**. Compared with observations, both the GWR-CF and GWR-CF-AOD methods

produce consistent long-term $R_s$ trends with SunDu-derived $R_s$ and CERES EBAF $R_s$

(**Figures 2, 3 and 9**), indicating that the GWR-CF and GWR-CF-AOD methods can

produce reasonable annual $R_s$ variations over China.

In zones I and II, located in northern arid/semiarid regions, the annual anomaly $R_s$

variation shows small fluctuations ranging from -10 to 10 W/m$^2$. In contrast, zones IV,

V, VIII and IX covering the Sichuan Basin, Yunnan-Guizhu Plateau, Qinghai-Tibet

Plateau and North China Plain show large $R_s$ variation trends. Li et al. (2018) found a

sharply increasing $R_s$ trend over East China, especially in the North China Plain, which
is due to controlling air pollution and reducing aerosol loading. However, our results
indicate that the increased surface solar radiation in North China is not confirmed by
satellite retrieval (CERES) and SunDu-derived $R_s$.

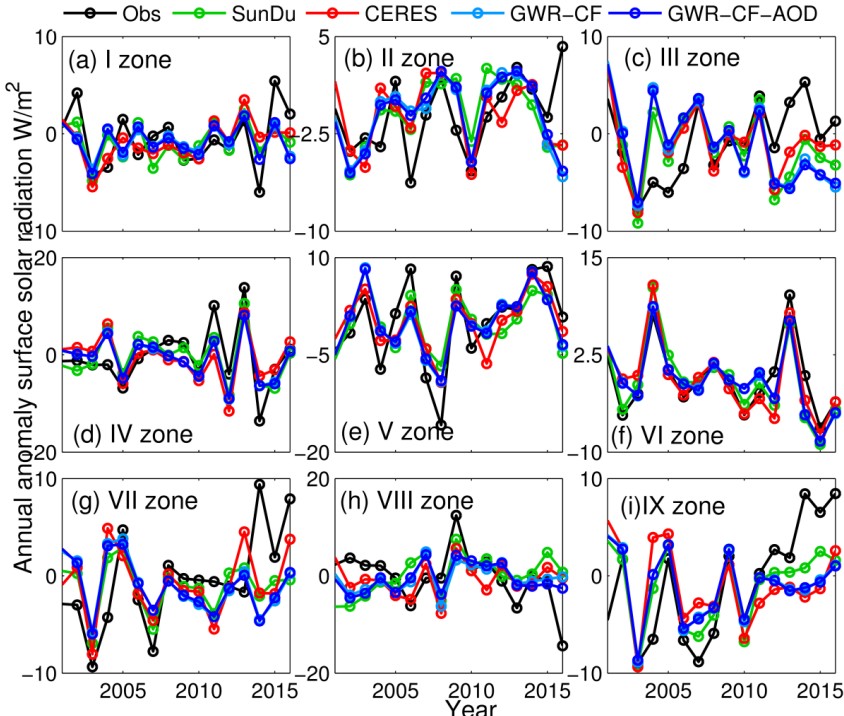


**Figure 9**. The regional mean of the annual anomaly of the surface solar radiation ($R_s$)

for different subregions. Nine subregions (I to IX) over China are shown in Figure 1.
Direct $R_s$ observations, SunDu-derived $R_s$, and CERES EBAF are shown as black lines,
green lines and red lines, respectively. Light and dark blue represent the $R_s$ variation
derived from the GWR-CF and the GWR-CF-AOD methods.

## 4. Discussion

### 4.1 Impact factors of $R_s$

In this study, we merged more than 2400 sunshine duration-derived $R_s$ site data
with MODIS CF and AOD data to generate high spatial resolution (0.1) $R_s$ over China
from 2000 to 2017. The results show that the GWR method incorporated with CF and
AOD (GWR-CF-AOD) performs best, indicating the non-neglected role of clouds and
aerosols in regulating the variation in $R_s$ over China.
Clouds and aerosols impact the solar radiation reaching the surface by radiative
absorption and scattering (Tang et al., 2017). Recent $R_s$ trend studies over Europe
suggest that CF may play a key role in the positive trend of $R_s$ since the 1990s (Pfeifroth
et al., 2018a). In terms of input data, our results also indicate that the cloud fraction
might be a major factor affecting $R_s$, which is consistent with our previous studies (Feng
and Wang, 2019).
Changes in aerosol loading have also been reported to be an important impact
factor (Che et al., 2005; Li et al., 2018; Liang and Xia, 2005; Qian et al., 2015; Xia,
2010; Zhou et al., 2019b). The atmospheric visibility data show that the slope of the
linear variation in surface solar radiation with respect to atmospheric visibility is
distinctly different at different stations (Yang et al., 2017), implying that the relationship
between $R_s$ and aerosols varies with location.

### 4.2 Performances of the fusion methods

The good overall performances of the GWR model have been reported in many
previous studies, including geography (Chao et al., 2018; Georganos et al., 2017),
economics (Ma and Gopal, 2018), meteorology (Li and Meng, 2017; Zhou et al., 2019a),
and epidemiology (Tsai and Teng, 2016). Chao et al. (2018) used the GWR method to
merge satellite precipitation and gauge observations to correct biases in satellite



precipitation data and downscale satellite precipitation to a finer spatial resolution at
the same time. Zhou et al. (2019a) used GWR to analyse haze pollution over China and
found that the GWR estimate was better than the OLS estimate, with an improvement
in correlation coefficient from 0.20 to 0.75.
Compared with other traditional interpolation methods, such as optimal
interpolation (OI), GWR can theoretically integrate geographical location and $R_s$ impact
factors for spatial $R_s$ estimations and reflect the non-stationary spatial relationship
between $R_s$ and its impact factors. The thin plate spline method can include CF and
AOD as covariates to simulate the approximately linear dependence of these impact
factors on $R_s$, but this linear function cannot fully describe the relationship among CF,
AOD and $R_s$ (Hong et al., 2005). Comparison results from Wang et al. (2017) also
indicate that the GWR method is better than the multiple linear regression method and
spline interpolation method for near surface air temperature.
**5. Data availability**
The merged $R_s$ product by GWR methods with cloud fraction and AOD data as
input in this study are available at https://doi.pangaea.de/10.1594/PANGAEA.921847
(Feng and Wang, 2020).
**6. Conclusions**
Accurate estimation of $R_s$ variability is crucially important for regional energy
budget, water cycle and climate change studies. Recent studies have shown that SunDu-
derived $R_s$ data can provide reliable long-term $R_s$ series. In this study, we merged
SunDu-derived Rs data with satellite-derived cloud fraction (CF) and aerosol optical
depth (AOD) data to generate high spatial resolution (0.1) $R_s$ over China from 2000 to



2017 (Feng and Wang, 2020). The GWR and OLS merging methods were also
compared.
Our results show that the spatial resolutions of all fusion results are improved to
0.1 °by incorporating MODIS cloud fraction data. The GWR shows better performance
than OLS, with increases in $R^2$ by 9.21%~12.81% and RMSEs reduced by
49.56%~54.68%, indicating that $R_s$ has complex characteristics of spatial variability
over China, which has also indicated the necessity of the high spatial resolution of $R_s$
data. As clouds and aerosols play vital roles in the variability in $R_s$, apparent
improvements in the results of SunDu-derived $R_s$ data merging are found if both cloud
fraction and AOD are incorporated. Based on the merging results incorporating only
cloud fraction, cloud fraction is suggested to be the major factor impacting $R_s$, which
explained approximately 86%~97% of $R_s$ variability. Generally, SunDu-derived $R_s$ data
merging results derived from GWR show more consistent multiyear mean $R_s$ and long-
term $R_s$ trends compared with those from OLS. Our results show that the improvement
in $R_s$ variability estimation is closely related to $R_s$ impact factors and $R_s$ spatial
heterogeneity. The merged $R_s$ products derived from GWR-CF-AOD can be
downloaded at https://doi.pangaea.de/10.1594/PANGAEA.921847. We also plan to
expand our $R_s$ dataset from 1983 to 2017 by using AVHRR based cloud retrievals.

## Acknowledgements

This study was funded by the National Key Research & Development Program of
China (2017YFA06036001), the National Natural Science Foundation of China
(41525018), the Fundamental Research Funds for the Central Universities
(#BLX201907), and the State Key Laboratory of Earth Surface Processes and Resource
Ecology (U2020-KF-02). We would like to thank Chengyang Xu, Yuna Mao, Jizeng





Du, Runze Li, Qian Ma, Guocan Wu, and Chunlue Zhou for their insightful comments.
We are grateful to Amelie Driemel for her help of uploading the data in PANGAEA.
The        cloud        data        can        be        downloaded        from
https://neo.sci.gsfc.nasa.gov/view.php?datasetId=MODAL2_M_CLD_FR.            The
CERES SYN data can be downloaded from https://ceres.larc.nasa.gov/data/.




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
