# Peer review of "Merging ground-based sunshine duration observations with satellite cloud and aerosol retrievals to produce high resolution long term surface solar radiation over China"

_Earth System Science Data, 2020_

## Referee Comment (RC1) · Anonymous Referee #1 · 22 Nov 2020

This is a nice work, in which SunDu-derived surface solar radiation (Rs) data are merged with satellite-derived cloud fraction and AOD data to generate high spatial resolution (0.1°) Rs over China. Both direct Rs observations (pyranometer data) at ~100 stations and sun duration records at 2400 stations are used in this study to demonstrate the reliable performances of the merging results. A striking result is that AOD plays a negligible role in the merging results, which indicates that the estimation method of Rs from sunshine measurements is robust and reliable. The result is valuable because long-term AOD retrievals are not accessible when building long-term Rs

data. The paper is well organized. I suggest to accept this submission after following issues are addressed. Major concerns: 1. It is not clear how to calculate clear sky Rs although a simple equation is given. A detailed introduction to the method is required since the conclusion mainly relies on the method. I wonder whether aerosol effect on Rs is accounted for by Sunshine duration measurement or by the equation used for the calculation of clear sky Rs. Addition, pls introduce more clearly which data are used in the calculation of clear sky Rs. 2. It was said that site dependent parameters were used in the equation 1 (e.q., a0-a2). I'm not sure how to derive these parameters at each station. 3. Frankly speaking, I'm not comfortable with the statement that the CERES EBAF can be taken as the reference. This seems based on the result that the agreement between SunDu-derived Rs and EBAR is much better than that between SunDu-derived Rs and pyranometer measurements. My opinion is that there seems possibility that aerosol effects were not properly accounted for by both SunDu-derived and satellite Rs algorithm. I mean this possibility cannot be fully eliminated, so it is suggestive to discuss this issue in somewhere.

Minor issues

1) Lines 31, 'Based on the SunDu-derived Rs from 97 meteorological observation stations. . .', the authors should mention that these 97 stations are co-located with those that direct Rs measurements sites. 2) Lines 130-133, what about the quality of the datasets from (Tang et al. 2019) and (Stengel et al. 2020)? I suggest the authors add detailed descriptions of these datasets. 3) Lines 164 to 165, the authors show that interpolation results have uncertainties due to the lack of detailed high spatial resolution information. What about the performances of machine learning methods in simulation of Rs. I suggest add more references here. 4) Line 178, "0.1" changes to "0.1°". 5) Lines 177 to 179, the authors merge the SunDu-derived Rs data with satellite-derived cloud fraction (CF) and AOD data. Why not directly merging the SunDu-derived Rs data with current Rs products? 6) Line 183,"sunDu" changes to "SunDu". 7) Add spatial resolution of each dataset and the references of each dataset in table 2. 8) Line

291, why not use MODIS AOD as input data in this study. 9) Lines316 to 317, SunDu derived Rs also contain the information of clouds, what about merging SunDu-derived Rs data only with AOD data? 10) Lines 390 to 392, two validation sites are randomly selected to evaluate the seasonal and annual variations in Rs. I suggest two sites with high AOD values and low AOD values. 11) Line 474, "0.1" changes to "0.1°". 12) Line 518, "0.1" changes to "0.1°". 13) Lines 535 to 536, deleted "We also plan to expand our Rs dataset from 1983 to 2017 by using AVHRR based cloud retrievals." Since this study focus the period from 2000 to 2016.

---

## Referee Comment (RC2) · Anonymous Referee #2 · 27 Nov 2020

This study attempts to generate a high resolution surface solar radiation (Rs) dataset. The idea is to construct a linear model between station based Rs, cloud fraction and AOD, and applies the model to the full study domain (China). While this dataset can be potentially useful, I don't understand how this approach could achieve a better accuracy than CERES 1 degree Rs product. This is because: (1) although the SunDu Rs can represent a much smaller area than the CERES 1 degree grid, SunDu Rs is validated using CERES Rs, which means that SunDu Rs cannot have a higher accuracy than CERES Rs, even at the 1 degree scale; (2) the AOD data used is still at 1 degree

resolution. This does not add much finer information and may be the reason why AOD has little impact on the prediction results. Overall, I don't see much value in this study unless the above question is addressed. Please see the specific comments below:

Major comments

1. The authors used SunDu Rs to train the model and to generate the high resolution Rs dataset. However, SunDu Rs is validated against CERES Rs, assuming that the latter has higher accuracy. On one hand, using grid based data to validate station based data is not appropriate. There can be a lot of variability within this 1 degree box. The authors did compare SunDu Rs with observed Rs but argued that their agreement is not as good as that between SunDu Rs and CERES Rs, and that the agreement between the latter two proofs the reliability of SunDu Rs. I don't agree with this argument. SunDu Rs should be directly validated against surface observed Rs. On the other hand, if CERES Rs is better than SunDu Rs, what's the point of using SunDu Rs to generate the 0.1 degree dataset? I guess using CERES Rs with 0.1 cloud and AOD would achieve at least the same accuracy, if not better. Yet, it has the advantage of full spatial coverage than SunDu Rs. 2. To proof the effect of fine resolution processing, a direct comparison with CERES should be provided. The authors can interpolate the CERES Rs to 0.1 degree and compare with their results. How difference are they? Are the differences physically explainable (i.e., related to cloud variability?).

Minor comments

1. What is the reason of the lower agreement between SunDu Rs and observed Rs? 2. Why using CERES 1dgree AOD? If spatial resolution matters, there are much finer products, such as the MODIS 1km and MODIS 3km products. 3. There are remote locations with very few SunDu stations, such as the Tibet plateau, are the relationships applicable? 4. It would be interesting to look at the spatial distribution of the coefficients. This can tell us some information about where clouds make a bigger impact and where aerosols are important. 5. What's the unit of Figure 2?

---

## Author Comment (AC1) · 4 Jan 2021

Reviewer #1 OVERALL 1) Comment: This is a nice work, in which SunDu-derived surface solar radiation (Rs) data are merged with satellite-derived cloud fraction and AOD data to generate high spatial resolution (0.1âŰę) Rs over China. Both direct Rs observations (pyranometer data) at âĹij100 stations and sun duration records at 2400 stations are used in this study to demonstrate the reliable performances of the merging results. A striking result is that AOD plays a negligible role in the merging results, which indicates that the estimation method of Rs from sunshine measurements is robust and

reliable. The result is valuable because long-term AOD retrievals are not accessible when building long-term Rs data. The paper is well organized. I suggest to accept this submission after following issues are addressed. Reply: The authors would like to thank anonymous referee #1 for his detailed and helpful comments. Below are our point by point responses to his comments.

GENERAL COMMENTS 2) Comment: It is not clear how to calculate clear sky Rs although a simple equation is given. A detailed introduction to the method is required since the conclusion mainly relies on the method. I wonder whether aerosol effect on Rs is accounted for by Sunshine duration measurement or by the equation used for the calculation of clear sky Rs. Addition, pls introduce more clearly which data are used in the calculation of clear sky Rs. Reply: For the clear sky Rs, $\tau$c_dir and $\tau$c_dif are calculated using a modified a broadband radiative transfer model by simplifying Leckner's spectral model (Leckner, 1978), which the effect of transmittance functions of permanent gas absorption, Rayleigh scattering, water vapour absorption, ozone absorption, and aerosol extinction are parameterized using the surface air temperature, surface pressure, precipitable water, the thickness of the ozone layer, turbidity as inputs (Yang et al., 2006). Calculation of Rs also includes impacts of aerosols because SunDu is impacted by changes in both clouds and aerosols (Wang, 2014).

3) Comment: It was said that site dependent parameters were used in the equation 1 (e.q., a0-a2). I'm not sure how to derive these parameters at each station. Reply: a0, a1, and a2 are the station-dependent parameters by tuning this equation with measurements of Rs and SunDu and then the method is applied regionally (Wang, 2014). Instead using observations from weather stations in Japan (Yang 2006), observations in CMA are used (Wang, 2014).

4) Comment: Frankly speaking, I'm not comfortable with the statement that the CERES EBAF can be taken as the reference. This seems based on the result that the agreement between SunDu-derived Rs and EBAR is much better than that between SunDu-derived Rs and pyranometer measurements. My opinion is that there seems possibility

that aerosol effects were not properly accounted for by both SunDu-derived and satellite Rs algorithm. I mean this possibility cannot be fully eliminated, so it is suggestive to discuss this issue in somewhere. Reply: Thanks for your suggestion, we agree with anonymous referee #1 that the data uncertainties cannot be fully eliminated for both ground observations and satellite retrievals. We will also discuss data uncertainties in the revised manuscript. The satellite Rs retrievals and SunDu derived Rs are generated by completely two different ways of measurements. Their correlation should be wake, but the high agreements of these two datasets from results indicate that CERES and SunDu-derived Rs can reflect the truth distribution of Rs in China to some extent. Similar results are also reported by (Wang et al., 2015) that SunDu-derived Rs have the best agreement with model-based Rs estimates, whereas satellite Rs retrievals, such as CERES, show best agreement with SunDu derived Rs and poor agreement with direct Rs observation due to the impact of thermal offset and directional response errors in direct observed Rs data. We notice that the biases of CERES EBAF are small. As mentioned in data section, the uncertainties of CERES EBAF data, reported by (Kato et al., 2018), in all sky global annual mean Rs is 4 W/m2. The SunDu data is a useful proxy of Rs, as mentioned in the data section. SunDu is almost free from influences of instrument replacement (Stanhill and Cohen, 2005). Even though, SunDu data do not provide a direct estimate of Rs and have the different sensitivity of atmospheric turbidity changes, compared with Rs observations, they are still a good proxy for variations of Rs (Manara et al., 2017). Moreover, existing studies have shown that SunDu-derived Rs estimates roughly depict long-term variability in Rs almost without the problems associated with early radiometry mentioned above (Wang, 2014; Wang et al., 2015).

MINOR ISSUES 5) Comment: Lines 31, 'Based on the SunDu-derived Rs from 97 meteorological observation stations: : :', the authors should mention that these 97 stations are co-located with those that direct Rs measurements sites. Reply: Thanks for your suggestion, we will add this information into the revised paper.

6) Comment: Lines 130-133, what about the quality of the datasets from (Tang et al. 2019) and (Stengel et al. 2020)? I suggest the authors add detailed descriptions of these datasets. Reply: Validation against the BSRN data indicated that SSR-tang have the mean bias error (MBE) of -11.5 W/m2 and root mean square error (RMSE) of 113.5 W/m2 for the instantaneous Rs estimates at 10 km scale, but (Tang et al., 2019) point out that care should be taken when using this dataset for trend analysis due to the absent of realistic aerosols input data. Stengel et al. (2020) also show that Rs derived from Cloud_cci AVHRR-PMv3 reveals a very good agreement against BSRN stations, with low standard deviations of 13.8 W/m2 and correlation coefficients above 0.98. While the bias for shortwave fluxes is small (1.9 W/m2). However, default an aerosol optical depth of 0.05 or data from Aerosol cci Level-2 or NASA MODIS Level-2 aerosol data are used in BUGSrad model to calculate clear sky Rs, indicating that impact of aerosols is not perfect parameterized in Cloud_cci AVHRR-PMv3. We will add this information into the revised paper.

7) Comment: Lines 164 to 165, the authors show that interpolation results have uncertainties due to the lack of detailed high spatial resolution information. What about the performances of machine learning methods in simulation of Rs. I suggest add more references here. Reply: The performances of different machine learning methods have been evaluated in many previous studies, including simulation Rs at regional scale with support of satellite retrievals (Wei et al., 2019; Yeom et al., 2019) and site scale by using routine meteorological observations (Cornejo-Bueno et al., 2019; Hou et al., 2020). Whatever models or training data are selected, the impacts of spatial relationship are not taken into account in these machine learning methods and therefore large number of input data are required to ensure accuracy. We will add this information into the revised paper.

8) Comment: Line 178, "0.1" changes to "0.1âŮę". Reply: We will correct it in the revised paper.

9) Comment: Lines 177 to 179, the authors merge the SunDu-derived Rs data with

satellite-derived cloud fraction (CF) and AOD data. Why not directly merging the SunDu-derived Rs data with current Rs products? Reply: We realize that we have not clearly explained this issue. Merging current Rs products with SunDu-derived Rs can also be applicable. Since many long-term Rs satellite products use climatology aerosols data as input, in this study, we want to know whether the merged product can achieve reliable Rs data without support of satellite derived aerosols input data.

10) Comment: Line 183,"sunDu" changes to "SunDu". Reply: We will correct it in the revised paper.

11) Comment: Add spatial resolution of each dataset and the references of each dataset in table 2. Reply: We will correct it in the revised paper.

12) Comment: Line 291, why not use MODIS AOD as input data in this study. Reply: We did not use MODIS AOD due to the impact of missing data. MODIS AOD conation missing values and can't meet the requirements of spatiotemporal continuity of AOD input in this study.

13) Comment: Lines316 to 317, SunDu derived Rs also contain the information of clouds, what about merging SunDu-derived Rs data only with AOD data? Reply: We agree with reviewers that SunDu derived Rs also contain the information of clouds. As cloud data can provide high resolution input data, we use cloud data to improve the spatial resolution of our merged data. We believe that with the support high resolution of AOD data, the merged data can produce more accurate results. However, the spatial resolution of current available AOD are 1 degree.

14) Comment: Lines 390 to 392, two validation sites are randomly selected to evaluate the seasonal and annual variations in Rs. I suggest two sites with high AOD values and low AOD values. Reply: We have checked the selected validation sites. The multiyear mean of AOD from Changchun and BeiHai are 0.49 and 0.70, respectively.

15) Comment: Line 474, "0.1" changes to "0.1âŮẹ". Reply: We will correct it in the

revised paper.

16) Comment: Line 518, "0.1" changes to "0.1âǓę". Reply: We will correct it in the revised paper.

17) Comment: Lines 535 to 536, deleted "We also plan to expand our Rs dataset from 1983 to 2017 by using AVHRR based cloud retrievals." Since this study focus the period from 2000 to 2016. Reply: We will correct it in the revised paper.

---

## Author Comment (AC2) · 4 Jan 2021

Reviewer #2 OVERALL 1) Comment: This study attempts to generate a high resolution surface solar radiation (Rs) dataset. The idea is to construct a linear model between station based Rs, cloud fraction and AOD, and applies the model to the full study domain (China). While this dataset can be potentially useful, I don't understand how this approach could achieve a better accuracy than CERES 1 degree Rs product. This is because: (1) although the SunDu Rs can represent a much smaller area than the CERES 1 degree grid, SunDu Rs is validated using CERES Rs, which means that

SunDu Rs cannot have a higher accuracy than CERES Rs, even at the 1 degree scale; (2) the AOD data used is still at 1 degree resolution. This does not add much finer information and may be the reason why AOD has little impact on the prediction results. Overall, I don't see much value in this study unless the above question is addressed. Please see the specific comments below: Reply: We realize that we have not clearly explained the significance of our work to generate high spatial resolution Rs data and the comparison results. We carefully think about all comments from anonymous referee #2. Below are our point by point responses to his comments.

MAJOR COMMENTS 2) Comment: The authors used SunDu Rs to train the model and to generate the high resolution Rs dataset. However, SunDu Rs is validated against CERES Rs, assuming that the latter has higher accuracy. On one hand, using grid based data to validate station based data is not appropriate. There can be a lot of variability within this 1 degree box. The authors did compare SunDu Rs with observed Rs but argued that their agreement is not as good as that between SunDu Rs and CERES Rs, and that the agreement between the latter two proofs the reliability of SunDu Rs. I don't agree with this argument. SunDu Rs should be directly validated against surface observed Rs. On the other hand, if CERES Rs is better than SunDu Rs, what's the point of using SunDu Rs to generate the 0.1 degree dataset? I guess using CERES Rs with 0.1 cloud and AOD would achieve at least the same accuracy, if not better. Yet, it has the advantage of full spatial coverage than SunDu Rs. Reply: We realize that we have not clearly explained the significance of our work and comparison results. In this study, we aim to build a reliable high resolution grid Rs data by establishing the physical spatial relationship between ground based SunDu derived Rs data with high resolution cloud satellite data with AOD to avoid the disadvantage of CERES for capturing the variability of Rs within a 1 degree box. The CERES and SunDu derived Rs are two completely different ways of measurements. Their correlation should be wake, but the high agreements of these two datasets from results indicate that CERES and SunDu-derived Rs can reflect the truth distribution of Rs to some extent. Similar results are also reported by (Wang et al., 2015) that SunDu-derived Rs have the best agreement
with model-based Rs estimates, whereas satellite Rs retrievals, such as CERES, show best agreement with SunDu derived Rs and poor agreement with direct Rs observation due to the impact of thermal offset and directional response errors in direct observed Rs data. We know that direct comparison between grid based data and station based data is not perfect. But direct comparison are widely used as a tradeoff way for validation in many studies due to lack of reliable high resolution grid Rs data. In this study, we aim to build this reliable high resolution grid Rs data. One may argue that using CERES Rs with 0.1 cloud and AOD can also produce high resolution Rs data. However, there are large amount of input data are require to ensure the accuracy of CERES. Most of these input data in CERES have low spatial resolution and limited spatial coverage and are only available after 2000. SunDu Rs have long time records with large spatial coverage. The merged SunDu derived Rs data can overcome these disadvantages of CERES and have the possibilities to build long term Rs by using AVHRR data.

3) Comment: To proof the effect of fine resolution processing, a direct comparison with CERES should be provided. The authors can interpolate the CERES Rs to 0.1 degree and compare with their results. How difference are they? Are the differences physically explainable (i.e., related to cloud variability?). Reply: Thanks for your suggestion. By using spatial interpolation method, CERES Rs can also be downscaled to 1km or 30m. These interpolated CERES Rs data cannot represent the detailed Rs distributions at spatial resolution of 1km or 30m. Without additional high spatial resolution data, interpolated cannot capture more detail variability of Rs. High spatial resolution cloud data can provide more detail information of cloud variability.

MINOR COMMENTS 4) Comment: What is the reason of the lower agreement between SunDu Rs and observed Rs? Reply: We realize that we have not clearly explained this issue. According to previous studies (Wang, 2014; Wang et al., 2015; Yang et al., 2018), the possible reasons of discrepancies between SunDu Rs and observed Rs are the changes in instrument and observation schedule of the observed Rs. (Wang, 2014; Wang et al., 2015; Yang et al., 2018) show that nearly half of observed

Rs (60 out of the 119 Rs observed stations) have inhomogeneity issues. These artificial changes points in observed Rs are mainly caused by instrument change (42 shifts), stations relocation (34 shifts), observation schedule change (20 shifts) and remaining 64 changepoints which could not be identified.

5) Comment: Why using CERES 1dgree AOD? If spatial resolution matters, there are much finer products, such as the MODIS 1km and MODIS 3km products. Reply: MODIS AOD conation missing values and can't meet the requirements of spatiotemporal continuity of AOD input in this study.

6) Comment: There are remote locations with very few SunDu stations, such as the Tibet plateau, are the relationships applicable? Reply: As shown in figure 9, the regional mean of the annual anomaly of the surface solar radiation (Rs) for zone II and zone VIII which are the regions such as the Tibet plateau. We notice that the merged Rs (GWR-CF-AOD) can produce consistent variation of Rs compared with observed data, indicating the relationships are applicable.

7) Comment: It would be interesting to look at the spatial distribution of the coefficients. This can tell us some information about where clouds make a bigger impact and where aerosols are important. Reply: According to the figure 6 in our previous study (Feng and Wang, 2019), cloud fraction shows strong negative correlation with Rs in most parts of China, while slight weak correlation coefficient near the north border of China. While clear sky Rs, which are primarily impact by the atmospheric aerosol loading, generally have small the correlation coefficient with Rs in most parts China.

8) Comment: What's the unit of Figure 2? Reply: The unit of Figure 2 is W/m2. We will add this information in the revised paper.

---

## Author Response (AR1)

**Reviewer #1**

OVERALL

1) **Comment:** This is a nice work, in which SunDu-derived surface solar radiation ($R_s$) data are merged with satellite-derived cloud fraction and AOD data to generate high spatial resolution (0.1∘) Rs over China. Both direct Rs observations (pyranometer data) at ~100 stations and sun duration records at 2400 stations are used in this study to demonstrate the reliable performances of the merging results. A striking result is that AOD plays a negligible role in the merging results, which indicates that the estimation method of Rs from sunshine measurements is robust and reliable. The result is valuable because long-term AOD retrievals are not accessible when building long-term Rs data. The paper is well organized. I suggest to accept this submission after following issues are addressed.

**Reply:** The authors would like to thank anonymous referee #1 for his detailed and helpful comments. Below are our point by point responses to his comments.

GENERAL COMMENTS

2) **Comment:** It is not clear how to calculate clear sky Rs although a simple equation is given. A detailed introduction to the method is required since the conclusion mainly relies on the method. I wonder whether aerosol effect on Rs is accounted for by Sunshine duration measurement or by the equation used for the calculation of clear sky Rs. Addition, pls introduce more clearly which data are used in the calculation of clear sky Rs.

**Reply:** Following anonymous referee #1 comments, we have added the description of the calculation of clear sky $R_s$ (**Lines 255-263**):

"For the clear sky $R_s$, $\tau_{c\_dir}$ and $\tau_{c\_dif}$ are calculated using a modified a broadband radiative transfer model by simplifying Leckner's spectral model (Leckner, 1978), which the effect of transmittance functions of permanent gas absorption, Rayleigh scattering, water vapour absorption, ozone absorption, and aerosol extinction are parameterized using the surface air temperature, surface pressure, precipitable water, the thickness of the ozone layer, turbidity as inputs (Yang et al., 2006). Calculation of $R_s$ also includes impacts of aerosols because SunDu is impacted by changes in both clouds and aerosols (Wang, 2014)."

Leckner, B. G.: The spectral distribution of solar radiation at the earth's surface— elements of a model, Sol. Energy, 20, 143-150, 1978.

Yang, K., Koike, T., and Ye, B.: Improving estimation of hourly, daily, and monthly solar radiation by importing global data sets, Agric. For. Meteorol., 137, 43-55, 2006.

Wang, K. C., Ma, Q., Li, Z., and Wang, J.: Decadal variability of surface incident solar radiation over China: Observations, satellite retrievals, and reanalyses, Journal of Geophysical Research Atmospheres, 120, 6500-6514, 2015.

3) **Comment:** It was said that site dependent parameters were used in the equation 1 (e.q., a0-a2). I'm not sure how to derive these parameters at each station.

**Reply:** We added the details of these site dependent parameters (**Lines 249-252**):

"$a_0$, $a_1$, and $a_2$ are the station-dependent parameters by tuning this equation with measurements of $R_s$ and SunDu and then the method is applied regionally (Wang, 2014). Instead using observations from weather stations in Japan (Yang et al., 2006), observations in CMA are used (Wang, 2014)."

Wang, K.C.: Measurement biases explain discrepancies between the observed and simulated decadal variability of surface incident solar radiation, Scientific Reports, 4, 6144, 2014.

Yang, K., Koike, T., and Ye, B.: Improving estimation of hourly, daily, and monthly solar radiation by importing global data sets, Agric. For. Meteorol., 137, 43-55, 2006.

4) **Comment:** Frankly speaking, I'm not comfortable with the statement that the CERES EBAF can be taken as the reference. This seems based on the result that the agreement between SunDu-derived Rs and EBAR is much better than that between SunDu-derived Rs and pyranometer measurements. My opinion is that there seems possibility that aerosol effects were not properly accounted for by both SunDu-derived and satellite Rs algorithm. I mean this possibility cannot be fully eliminated, so it is suggestive to discuss this issue in somewhere.

**Reply:** Thanks for your suggestion, we agree with anonymous referee #1 that the data uncertainties cannot be fully eliminated for both ground observations and satellite retrievals. As mentioned in data section, the uncertainties of CERES EBAF data, reported by (Kato et al., 2018), in all sky global annual mean $R_s$ is 4 W/m$^2$. The descriptions of uncertainties of SunDu derived $R_s$ are added (**Lines 84-87**). The satellite $R_s$ retrievals and SunDu derived $R_s$ are totally independent, but the high agreements of these two datasets indicate that they both are of higher accuracy. We will also discuss this issue in the revised manuscript (**Lines 269-273**):

"Even though, SunDu data do not provide a direct estimate of $R_s$ and have the different sensitivity of atmospheric turbidity changes, compared with $R_s$ observations, they are still a good proxy for variations of $R_s$ (Manara et al., 2017)."

"The satellite $R_s$ retrievals and SunDu derived $R_s$ are totally independent, but the high agreements of these two datasets indicate that they both are of higher accuracy. Similar results are also reported by (Wang et al., 2015) that low agreement between SunDu derived $R_s$ and direct $R_s$ observation is likely due to the directional response errors of the direct observations of $R_s$."

Kato, S., Rose, F. G., Rutan, D. A., Thorsen, T. J., Loeb, N. G., Doelling, D. R., Huang, X., Smith, W. L., Su, W., and Ham, S.-H.: Surface Irradiances of Edition 4.0 Clouds and the Earth's Radiant Energy System (CERES) Energy Balanced and Filled (EBAF) Data Product, J. Clim., 31, 4501-4527, 2018.

Manara, V., Brunetti, M., Maugeri, M., Sanchez-Lorenzo, A., and Wild, M.: Sunshine duration and global radiation trends in Italy (1959–2013): To what extent do they agree?, journal of geophysical research, 122, 4312-4331, 2017.

Wang, K.C.: Measurement biases explain discrepancies between the observed and simulated decadal variability of surface incident solar radiation, Scientific Reports, 4, 6144, 2014.

MINOR ISSUES

5) **Comment:** Lines 31, 'Based on the SunDu-derived Rs from 97 meteorological observation stations: : :', the authors should mention that these 97 stations are co-located with those that direct Rs measurements sites.

**Reply:** Thanks for your suggestion, we will add this information in **Lines 31-32**:

"Based on the SunDu-derived $R_s$ from 97 meteorological observation stations, which are co-located with those that direct $R_s$ measurement sites…"

6) **Comment:** Lines 130-133, what about the quality of the datasets from (Tang et al. 2019) and (Stengel et al. 2020)? I suggest the authors add detailed descriptions of these datasets.

**Reply:** we added the description of these dataset (**Lines 136-146**):

"Validation against the BSRN data indicated that SSR-tang have the mean bias error (MBE) of -11.5 W/m$^2$ and root mean square error (RMSE) of 113.5 W/m$^2$ for the instantaneous $R_s$ estimates at 10 km scale, but (Tang et al., 2019) point out that care should be taken when using this dataset for trend analysis due to the absent of realistic aerosols input data. Stengel et al. (2020) also show that $R_s$ derived from Cloud_cci AVHRR-PMv3 reveals a very good agreement against BSRN stations, with low standard deviations of 13.8 W/m$^2$ and correlation coefficients above 0.98. While the bias for shortwave fluxes is small (1.9 W/m$^2$). However, default an aerosol optical depth of 0.05 or data from Aerosol cci Level-2 or NASA MODIS Level-2 aerosol data are used in BUGSrad model to calculate clear sky $R_s$, indicating that impact of aerosols is not perfect parameterized in Cloud_cci AVHRR-PMv3."

Tang, W., Yang, K., Qin, J., Li, X., and Niu, X.: A 16-year dataset (2000–2015) of high-resolution (3 h, 10 km) global surface solar radiation, Earth Syst. Sci. Data, 11, 1905-1915, 2019.

Stengel, M., Stapelberg, S., Sus, O., Finkensieper, S., Würzler, B., Philipp, D., Hollmann, R., Poulsen, C., Christensen, M., and McGarragh, G.: Cloud_cci Advanced Very High Resolution Radiometer post meridiem (AVHRR-PM) dataset version 3: 35-year climatology of global cloud and radiation properties, Earth Syst. Sci. Data, 12, 41-60, 2020.

7) **Comment:** Lines 164 to 165, the authors show that interpolation results have uncertainties due to the lack of detailed high spatial resolution information. What about the performances of machine learning methods in simulation of Rs. I suggest add more references here.

**Reply:** we added the description of these dataset (**Lines 183-189**):

"The performances of different machine learning methods have been evaluated in many previous studies, including simulation $R_s$ at regional scale with support of satellite retrievals (Wei et al., 2019; Yeom et al., 2019) and site scale by using routine meteorological observations (Cornejo-Bueno et al., 2019; Hou et al., 2020). Whatever models or training data are selected, the impacts of spatial relationship are not taken into account in these machine learning methods and therefore large number of input data are required to ensure accuracy."

Wei, Y., Zhang, X., Hou, N., Zhang, W., Jia, K., and Yao, Y.: Estimation of surface downward shortwave radiation over China from AVHRR data based on four machine learning methods, Sol. Energy, 177, 32-46, 2019.

Yeom, J. M., Park, S., Chae, T., Kim, J. Y., and Lee, C. S.: Spatial Assessment of Solar Radiation by Machine Learning and Deep Neural Network Models Using Data Provided by the COMS MI Geostationary Satellite: A Case Study in South Korea, Sensors (Basel), 19, 2019.

Cornejo-Bueno, L., Casanova-Mateo, C., Sanz-Justo, J., and Salcedo-Sanz, S.: Machine learning regressors for solar radiation estimation from satellite data, Sol. Energy, 183, 768-775, 2019.

Hou, N., Zhang, X., Zhang, W., Xu, J., Feng, C., Yang, S., Jia, K., Yao, Y., Cheng, J., and Jiang, B.: A New Long-Term Downward Surface Solar Radiation Dataset over China from 1958 to 2015, Sensors (Basel), 20, 2020.

8) **Comment:** Line 178, "0.1" changes to "0.1∘".

**Reply:** Corrected as suggestions (**Line 199**).

9) **Comment:** Lines 177 to 179, the authors merge the SunDu-derived Rs data with satellite-derived cloud fraction (CF) and AOD data. Why not directly merging the SunDu-derived Rs data with current Rs products?

**Reply:** Merging current Rs products with SunDu-derived $R_s$ can also be applicable. Since many long-term $R_s$ satellite products use climatology aerosols data as input, in this study, we want to know whether the merged product can achieve reliable $R_s$ data without support of satellite derived aerosols input data.

10) **Comment:** Line 183,"sunDu" changes to "SunDu".

**Reply:** We have deleted the sentence. (**Line 209**)

11) **Comment:** Add spatial resolution of each dataset and the references of each dataset in table 2.

**Reply:** Corrected as suggestions (**Line 238**).

12) **Comment:** Line 291, why not use MODIS AOD as input data in this study.

**Reply:** we added the description of why we did not use MODIS AOD as input (Lines **318-322**):

"We did not use AOD from MODIS, because MODIS AOD conation missing values and can't meet the requirements of spatiotemporal continuity of AOD input in this study. In addition, MODIS AOD is only available under clear sky conditions while AOD provided by the assimilation system is averaged under all conditions."

13) **Comment:** Lines316 to 317, SunDu derived Rs also contain the information of clouds, what about merging SunDu-derived Rs data only with AOD data?

**Reply:** We agree with reviewers that SunDu derived $R_s$ also contain the information of clouds. As cloud data can provide high resolution input data, we use cloud data to improve the spatial resolution of our merged data. We believe that with the support high resolution of AOD data, the merged data can produce more accurate results. However, the spatial resolution of current available AOD are 1 degree.

14) **Comment:** Lines 390 to 392, two validation sites are randomly selected to evaluate the seasonal and annual variations in Rs. I suggest two sites with high AOD values and low AOD values.

**Reply:** We have checked the selected validation sites. We added this information in the revised manuscript (**Lines 422-423 and Lines 437-438**).

"The multiyear mean of AOD from Changchun and BeiHai are 0.49 and 0.70, respectively."

15) **Comment:** Line 474, "0.1" changes to "0.1∘".

**Reply:** Corrected as suggestions (**Line 505**).

16) **Comment:** Line 518, "0.1" changes to "0.1∘".

**Reply:** Corrected as suggestions (**Line 555**).

17) **Comment:** Lines 535 to 536, deleted "We also plan to expand our Rs dataset from 1983 to 2017 by using AVHRR based cloud retrievals." Since this study focus the period from 2000 to 2016.

**Reply:** Corrected as suggestions.

OVERALL

1) **Comment:** This study attempts to generate a high resolution surface solar radiation (Rs) dataset. The idea is to construct a linear model between station based Rs, cloud fraction and AOD, and applies the model to the full study domain (China). While this dataset can be potentially useful, I don't understand how this approach could achieve a better accuracy than CERES 1 degree Rs product. This is because: (1) although the SunDu Rs can represent a much smaller area than the CERES 1 degree grid, SunDu Rs is validated using CERES Rs, which means that SunDu Rs cannot have a higher accuracy than CERES Rs, even at the 1 degree scale; (2) the AOD data used is still at 1 degree resolution. This does not add much finer information and may be the reason why AOD has little impact on the prediction results. Overall, I don't see much value in this study unless the above question is addressed. Please see the specific comments below:

**Reply:** We realize that we have not clearly explained the significance of our work to generate high spatial resolution $R_s$ data and the comparison results. We carefully think about all comments from anonymous referee #2. Below are our point by point responses to the comments.

MAJOR COMMENTS

2) **Comment:** The authors used SunDu Rs to train the model and to generate the high resolution Rs dataset. However, SunDu Rs is validated against CERES Rs, assuming that the latter has higher accuracy. On one hand, using grid based data to validate station based data is not appropriate. There can be a lot of variability within this 1 degree box. The authors did compare SunDu Rs with observed Rs but argued that their agreement is not as good as that between SunDu Rs and CERES Rs, and that the agreement between the latter two proofs the reliability of SunDu Rs. I don't agree with this argument. SunDu Rs should be directly validated against surface observed Rs. On the other hand, if CERES Rs is better than SunDu Rs, what's the point of using SunDu Rs to generate the 0.1 degree dataset? I guess using CERES Rs with 0.1 cloud and AOD would achieve at least the same accuracy, if not better. Yet, it has the advantage of full spatial coverage than SunDu Rs.

**Reply:** We realize that we have not clearly explained the significance of our work and comparison results. In this study, we aim to build a reliable high resolution grid $R_s$ data by establishing the physical spatial relationship between ground based SunDu derived $R_s$ data with high resolution cloud satellite data with AOD to avoid the disadvantage of CERES for capturing the variability of $R_s$ within a 1 degree box. We have refined the description of our goals in the end of introduction (**Lines 205-209**): "Since current $R_s$ high quality $R_s$ such as CERES EBAF have low spatial resolution, the output of this study provides a reliable high resolution grid $R_s$ data to avoid the disadvantage of CERES EBAF for capturing the variability of $R_s$ within a 1 degree box and provide guidance to merge multisource data to generate long-term $R_s$ data over China." We know that direct comparison between grid based data and station based data is not perfect. "However we show that the satellite $R_s$ retrievals and SunDu derived $R_s$ are totally independent, but the high agreements of these two datasets indicate that they both are of higher accuracy. Similar results are also reported by (Wang et al., 2015) that low agreement between SunDu derived $R_s$ and direct $R_s$ observation is likely due to the directional response errors of the direct observations of $R_s$" (**Lines 270-273**). We know that direct comparison between grid based data and station based data is not perfect. But direct comparison are widely used as a tradeoff way for validation in many studies due to lack of reliable high resolution grid $R_s$ data. In this study, we aim to build this reliable high resolution grid $R_s$ data. One may argue that using CERES $R_s$ with 0.1 cloud and AOD can also produce high resolution $R_s$ data. However, there are large amount of input data are require to ensure the accuracy of CERES. Most of these input data in CERES have low spatial resolution and limited spatial coverage and are only available after 2000. SunDu $R_s$ have long time records with large spatial coverage. The merged SunDu derived $R_s$ data can overcome these disadvantages of CERES and have the possibilities to build long term $R_s$ by using AVHRR data.

3) **Comment:** To proof the effect of fine resolution processing, a direct comparison with CERES should be provided. The authors can interpolate the CERES Rs to 0.1 degree and compare with their results. How difference are they? Are the differences physically explainable (i.e., related to cloud variability?).

**Reply:** Thanks for your suggestion. We have discussed this issue in the discussion section (Lines 542-545)

"By using spatial interpolation method, CERES $R_s$ can also be downscaled to 1km or 30m. These interpolated CERES $R_s$ data cannot represent the detailed $R_s$ distributions at spatial resolution of 1km or 30m. Without additional high spatial resolution data, interpolated cannot capture more detail variability of $R_s$. High spatial resolution cloud data can provide more detail information of cloud variability."

MINOR COMMENTS

4) **Comment:** What is the reason of the lower agreement between SunDu Rs and observed Rs?

**Reply:** The reason of the lower agreement between SunDu Rs and observed Rs have added in the revised manuscript (**Lines 271-273**)

"Similar results are also reported by (Wang et al., 2015) that low agreement between SunDu derived $R_s$ and direct $R_s$ observation is likely due to the directional response errors of the direct observations of $R_s$."

Wang, K. C., Ma, Q., Li, Z., and Wang, J.: Decadal variability of surface incident solar radiation over China: Observations, satellite retrievals, and reanalyses, Journal of Geophysical Research Atmospheres, 120, 6500-6514, 2015.

5) **Comment:** Why using CERES 1dgree AOD? If spatial resolution matters, there are much finer products, such as the MODIS 1km and MODIS 3km products.

**Reply:** We have added the reasons in (**Lines 319-323**):

"We did not use AOD from MODIS, because MODIS AOD conation missing values and can't meet the requirements of spatiotemporal continuity of AOD input in this study. In addition, MODIS AOD is only available under clear sky conditions while AOD provided by the assimilation system is averaged under all conditions."

6) **Comment:** There are remote locations with very few SunDu stations, such as the

Tibet plateau, are the relationships applicable?

**Reply:** As shown in figure 9, the regional mean of the annual anomaly of the surface solar radiation ($R_s$) for zone II and zone VIII which are the regions such as the Tibet plateau. We notice that the merged $R_s$ (GWR-CF-AOD) can produce consistent variation of Rs compared with observed data, indicating the relationships are applicable.

7) **Comment:** It would be interesting to look at the spatial distribution of the coefficients. This can tell us some information about where clouds make a bigger impact and where aerosols are important.

**Reply:** According to the figure 6 in our previous study (Feng and Wang, 2019), cloud fraction shows strong negative correlation with $R_s$ in most parts of China, while slight weak correlation coefficient near the north border of China. While clear sky $R_s$, which are primarily impact by the atmospheric aerosol loading, generally have small the correlation coefficient with $R_s$ in most parts China.

Feng, F. and Wang, K.: Determining Factors of Monthly to Decadal Variability in Surface Solar Radiation in China: Evidences from Current Reanalyses, J. Geophys. Res. Atmos., 124, 9161-9182, 2019.

8) **Comment:** What's the unit of Figure 2?

**Reply:** We have added this information in the revised paper (**Lines 286-287**):

[revised manuscript text omitted]

---

## Author Response (AR2)

**Reviewer**

**OVERALL**

1) **Comment:** The authors need to prove that SunDu Rs can add value to the 0.1 degree product, instead of cloud fraction data alone. I therefore suggest the authors perform a similar regression (GWR) analysis but using CERES data interpolated to 0.1 degree and 0.1 degree cloud, and compare the results with those using SunDu Rs.

**Reply:** The authors would like to thank anonymous reviewers for their valuable and constructive comments, which help us to further improve the manuscript. According to the reviewers' comments, we perform a similar GWR analysis but using CERES data interpolated to 0.1 degree and 0.1 degree cloud, and compare the results with those using SunDu  $R_s$ . As the subplots (c) and (d) of figure 1S shown, both data use same 0.1 degree cloud as auxiliary data to perform the GWR analysis. The differences are that GWR CF  $R_s$  are based on SunDu derived  $R_s$ , while GWR CERES CF 0.1 degree are based on the CERES data which interpolated to 0.1 degree. This results suggest that SunDu  $
[revised manuscript text omitted]